# FINE-TUNING ATTENTION MODULES ONLY: ENHANCING WEIGHT DISENTANGLEMENT IN TASK ARITHMETIC

**Ruochen Jin**[*,1,2]**, Bojian Hou**[*,2]**, Jiancong Xiao**[*,2]**, Weijie J. Su**[2]**, and Li Shen**[2]
[1] East China Normal University, Shanghai, China
[2] University of Pennsylvania, Philadelphia, PA, USA
{ruochenj,bojianh,jcxiao}@upenn.edu, suw@wharton.upenn.edu,
lishen@upenn.edu

## ABSTRACT

In recent years, *task arithmetic* has garnered increasing attention. This approach edits pre-trained models directly in weight space by combining the fine-tuned weights of various tasks into a *unified model*. Its efficiency and cost-effectiveness stem from its training-free combination, contrasting with traditional methods that require model training on large datasets for multiple tasks. However, applying such a unified model to individual tasks can lead to interference from other tasks (lack of *weight disentanglement*). To address this issue, Neural Tangent Kernel (NTK) linearization has been employed to leverage a "kernel behavior", facilitating weight disentanglement and mitigating adverse effects from unrelated tasks. Despite its benefits, NTK linearization presents drawbacks, including doubled training costs, as well as reduced performance of individual models. To tackle this problem, we propose a simple yet effective and efficient method that is to finetune the attention modules only in the Transformer. Our study reveals that the attention modules exhibit kernel behavior, and fine-tuning the attention modules only significantly improves weight disentanglement. To further understand how our method improves the weight disentanglement of task arithmetic, we present a comprehensive study of task arithmetic by differentiating the role of the representation module and task-specific module. In particular, we find that the representation module plays an important role in improving weight disentanglement whereas the task-specific modules such as the classification heads can degrade the weight disentanglement performance. [1]

## 1 INTRODUCTION

The emergence of large pre-trained models in the open-source community has significantly expanded the potential to enhance performance on downstream tasks (Ilharco et al., 2023; 2022; Zhuang et al., 2020), align with human preferences (Lu et al., 2022; Glaese et al., 2022; Xiao et al., 2024; Li et al., 2025; Wang et al., 2025), and improve robustness (Hou et al., 2017; Ortiz-Jiménez et al., 2021; Santurkar et al., 2021; Tancik et al., 2020). However, traditional methods often involve expensive joint fine-tuning across multiple tasks (Zhuang et al., 2020) and rely heavily on human feedback (Ouyang et al., 2022), which limits their scalability and broad adoption. Moreover, optimizing performance for specific downstream tasks usually compromises the model's initial pre-training performance or zero-shot accuracy (French, 1999; McCloskey & Cohen, 1989).

In light of these challenges, the necessity of task arithmetic in multitask learning has become increasingly evident. *Task arithmetic* offers a cost-effective and efficient alternative by enabling training-free combinations in the weight space of pre-trained models without sacrificing the model's original capabilities (Ilharco et al., 2023). Central to this approach is the concept of a *task vector*, which represents a set of weight adjustments specifically calibrated for a given task through fine-tuning,

---

[*]Equal Contribution
[1]The code is available at https://github.com/kyrie-23/task_arithmetic_tangent.

Table 1: **Task arithmetic performance comparison between different methods.** The task arithmetic performance (the average accuracy of the unified model over all the tasks) of our method outperforms the state-of-the-art due to both good performance of individual models and good kernel behavior (weight disentanglement).

|  | Performance of Individual Models | Kernel Behavior (Weight Disentanglement) | Task Arithmetic Performance |
|---|---|---|---|
| Ilharco et al. (2023) | ✓ | - | 70.00% |
| Ortiz-Jimenez et al. (2024) | - | ✓ | 76.26% |
| **Ours** | ✓ | ✓ | 78.37% |

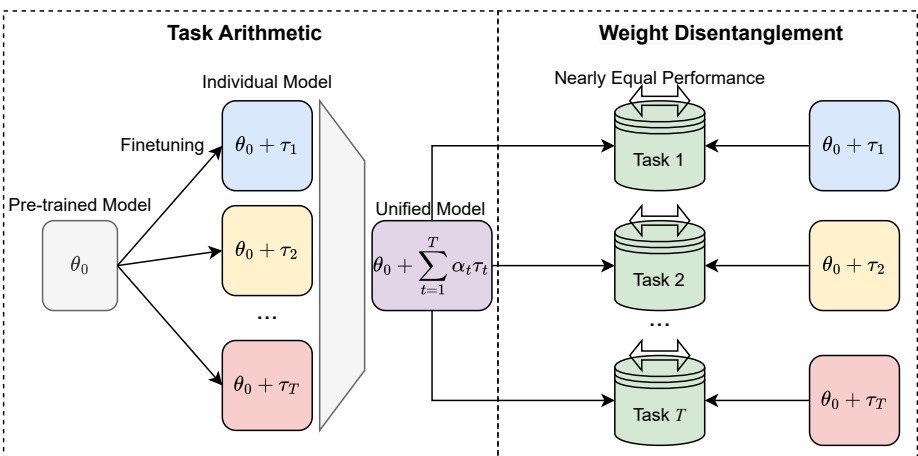

Figure 1: **Illustration of the concepts of task arithmetic and weight disentanglement.** On the left-hand side, in task arithmetic, we first finetune the pre-trained model $\theta_0$ and get the finetuned individual model $\theta_0 + \alpha_t \tau_t$ where $\tau_t$ is the $t$th task vector. We eventually obtain the unified model by adding all the task vectors to the pre-trained model: $\theta_0 + \sum_{t=1}^{T} \alpha_t \tau_t$. On the right-hand side, weight disentanglement means that the prediction of the unified model on a specific task will not be affected by other tasks.

obtained by subtracting the task-specific weights from the original pre-trained weights (Ilharco et al., 2023). Each task vector encodes a unique representational signature tailored to a particular task.

As illustrated in Figure 1 (left), $\theta_0$ is the pretrained model and $\tau_t$, $t = 1, \cdots, T$ is the $t$th task vector. The *individual model* on each task is derived by $\theta_0 + \alpha_t \tau_t$. Task arithmetic is to add all the task vectors to the pre-trained model to obtain the *unified model* $\theta_0 + \sum_{t=1}^{T} \alpha_t \tau_t$. A main goal in task arithmetic is to achieve weight disentanglement as shown in Figure 1 (right) which means the prediction performance of the unified model such as the accuracy on a specific task will not be affected by other tasks. In other words, the unified model has nearly equal performance with the individual model.

The reason that we aim to achieve weight disentanglement is because task arithmetic primarily focuses on competing tasks rather than synergistic ones (Ilharco et al., 2023), as the challenge lies in balancing and optimizing performance across tasks that may have conflicts or require different model behaviors. This emphasis on competing tasks is crucial for developing robust multi-task models that can effectively handle a diverse range of applications without performance degradation.

However, weight disentanglement remains the most formidable challenge in task arithmetic. Recent research (Ortiz-Jimenez et al., 2024) has shown that constraining models to fine-tune within the *tangent space* significantly improves weight disentanglement, thanks to the inherent *kernel behavior* of models during early fine-tuning stages. This kernel behavior, formalized by the Neural Tangent Kernel (NTK) theory (Jacot et al., 2018), refers to neural networks updating primarily around pre-trained parameter initializations. While NTK linearization is effective, it compromises the performance of

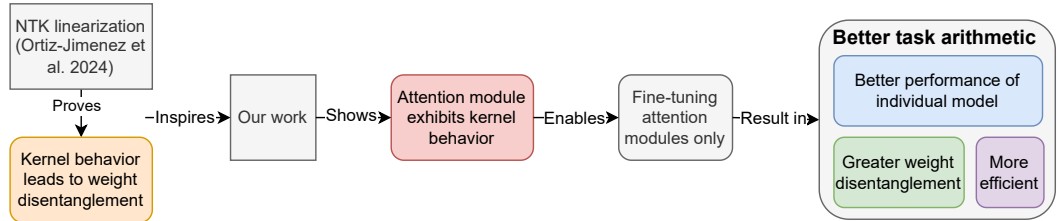

Figure 2: Logic flow of our work.

individual models and demands two to three times more computational resources, conflicting with task arithmetic's original efficiency goals.

To tackle this problem, we hypothesize that a combination of non-linear fine-tuning and weight disentanglement is necessary. This hypothesis motivates us to investigate the following question: Does there exist a sub-module within neural networks where non-linear fine-tuning can exhibit weight disentanglement? We demonstrate that attention modules in transformer models exhibit kernel behavior (see Figure 3) which is crucial for weight disentanglement. This finding is supported by empirical evidence using post-hoc linearization (Ortiz-Jimenez et al., 2024), a technique that approximates the change in network output after training using a first-order Taylor expansion. Building upon the insights above, we propose to fine-tune the attention modules only, which will improve the performance of individual models and efficiency while maintaining the weight disentanglement. This logic flow is illustrated in Figure 2.

By focusing on fine-tuning only the attention modules, our approach significantly improves weight disentanglement and accuracy of the unified model compared to non-linear fine-tuning and NTK linearization (see Table 2), while substantially reducing computational burden and memory usage. This method offers a balanced alternative that maintains strong performance of individual models while enhancing weight disentanglement capabilities, providing a practical solution for improving task arithmetic performance without sacrificing efficiency or accuracy.

To further understand how our method improves the weight disentanglement of task arithmetic, we present a study by differentiating the role of the representation module and task-specific module, while existing literature (Ortiz-Jimenez et al., 2024) formulated task arithmetic using a single model without clearly differentiating them. We conduct a comprehensive study of task arithmetic on pre-trained Vision Transformer (ViT) models like the Contrastive Language-Image Pre-Training (CLIP) (Radford et al., 2021), providing new insights into its fundamental mechanisms and proposing novel methods to improve the performance of pre-trained models through task arithmetic. Specifically, we illustrate that the representation module plays an important role in improving weight disentanglement whereas this has been constrained by task-specific modules, such as classification heads.

In particular, our main contributions are as follows:

- We propose a simple yet effective and efficient method that only fine-tunes attention modules, which improves weight disentanglement and the average accuracy of the unified model on all the tasks up to 2.38% improvement compared to the state-of-the-art methods and 8.37% over the non-linear baseline on several vision-language benchmarks.
- We demonstrate that the attention module exhibits kernel behavior, suggesting that focusing on fine-tuning these modules could enhance the weight disentanglement capabilities in task arithmetic while maintaining efficiency.
- We reformulate the architecture of task arithmetic by separating the representation module from task-specific modules, revealing that while weight disentanglement mostly comes from the representation module, the effectiveness of task arithmetic is constrained by task-specific components like classification heads.

## 2 PRELIMINARIES: TASK ARITHMETIC AND WEIGHT DISENTANGLEMENT

We begin by introducing the necessary mathematical notations. Let $F : \mathcal{X} \times \Theta \to \mathcal{Y}$ be a neural network taking inputs $x \in \mathcal{X}$ and parameterized by a set of weights $\vartheta \in \Theta$, which consists of a

representation module $f(\cdot; \theta)$ and a task-specific module $g(\cdot; \omega)$ where $\vartheta = \{\theta, \omega\}$. We assume $\mathcal{X} \subseteq \mathbb{R}^d$, $\Theta \subseteq \mathbb{R}^m$ and $\mathcal{Y} \subseteq \mathbb{R}^c$. Consider $T$ tasks, with every task $t$ consisting of a triplet $(D_t, \mu_t, F_t^*)$, where $D_t \subseteq \mathcal{X}$ is a data support (e.g., ImageNet (Deng et al., 2009) images), $\mu_t$ an input distribution such that $\text{supp}(\mu_t) = D_t$, and $F_t^* : D_t \to \mathcal{Y}$ a target function (e.g., labels). In practice, each task is identified with a training set $\{(x_v, F_t^*(x_v))\}_{v \in [n]}$ where $F_t^*(x_v) = g(f(x_v; \theta_t^*); \omega_t)$ with $x \sim \mu_t$, that is used to fine-tune the representation modules starting from the pre-trained weights $\theta_0$ and to obtain the fine-tuned weights $\theta_t^*$, while the task-specific modules are fixed at $\omega_t$.

Having established this context, we can now introduce the key concepts of this work: task vectors and task arithmetic.

**Definition 1 (Task Vector and Task Arithmetic)** *Let $\theta_0$ denote the parameters of the pre-trained representation modules and $\theta_t$ denote the parameters after fine-tuning on task $t$. The task vector $\tau_t$ for task $t$ is defined as: $\tau_t = \theta_t - \theta_0$. Task arithmetic is an operation of adapting a pre-trained model to $T$ different tasks by modifying the pre-trained parameters $\theta_0$ to the unified parameters $\theta_{\text{unified}}$ as follows:*

$$\theta_{\text{unified}} = \theta_0 + \sum_{t=1}^{T} \alpha_t \tau_t,$$

*where $\alpha_t$ are scalar coefficients.*

To distinguish from the concept of a *unified model*, we refer to the model $f_{\theta_t}$ fine-tuned on a specific task $T$ as an *individual model*. It is worth noting that adding a single task vector $\tau_t$ to a pre-trained model with a coefficient $\alpha = 1$ yields a model equivalent to this individual model $f_{\theta_t}$.

**Non-linear Fine-Tuning.** The initial idea of task arithmetic was introduced by Ilharco et al. (2023). They demonstrated that performance of the unified model on multiple tasks could be improved simultaneously through task arithmetic. However, the performance of the model still lagged behind that of individual models specifically fine-tuned for particular tasks. In the remainder of this paper, we refer to this baseline method as *non-linear fine-tuning*, as the task vectors $\tau_t$ are obtained through this approach.

**Accuracy Gap.** We define the *accuracy gap* as the difference in accuracy on task $t$ between the unified model and the corresponding individual model. A plausible hypothesis for this accuracy gap is that the task vectors for different tasks exhibit implicit conflicts with one another. To address the limitations of non-linear fine-tuning, Ortiz-Jimenez et al. (2024) proposed that *weight disentanglement* is an important and potentially necessary condition for effective task arithmetic.

**Definition 2 (Weight disentanglement)** *Given $T$ different tasks and their corresponding supports $D = \{D_t\}_{t \in [T]}$. We say a set of task vectors $\mathcal{T} = \{\tau_t\}_{t \in [T]}$ is weight disentangled with respect to a parametric function $f : \mathcal{X} \times \Theta \to \mathcal{Y}$ and the initial weights $\theta_0$, if*

$$f\left(x; \theta_0 + \sum_{t=1}^{T} \alpha_t \tau_t\right) = \sum_{t=1}^{T} f(x; \theta_0 + \alpha_t \tau_t) \mathbb{1}(x \in D_t) + f(x; \theta_0) \mathbb{1}\left(x \notin \bigcup_{t \in [T]} D_t\right). \quad (1)$$

*where $f$ is the representation module.*

**Remark.** Although the concept of weight disentanglement was originally proposed by Ortiz-Jimenez et al. (2024), our Definition 2 differs from theirs in several key aspects.

Firstly, our definition characterizes weight disentanglement as a property of a set of task vectors w.r.t. the function $f$, whereas it was originally defined as a property of the function $f$ w.r.t. the task vectors. There are two reasons for our reversed formulation. 1) The term "weight disentanglement" corresponds to $T$ different weights (i.e., task vectors), rather than $f$. 2) Both Ortiz-Jimenez et al. (2024)'s and our approach aim to find better task vectors $\mathcal{T}$, rather than finding better $f$.

Secondly, our definition applies specifically to the representation module, whereas the original definition encompasses the entire neural network. This is primarily motivated by the fact that task arithmetic is performed exclusively on the representation module. We will show later in Section 4 that this definition is well-defined: weight disentanglement emerges from the representation module.

**NTK Linearization Fine-tuning.** Inspired by studies of NTK showing that very wide neural networks behave similarly to linear functions around their initialization $\theta_0$, Ortiz-Jiménez et al. (2021) proposed fine-tuning the task vectors $\tau$ on a post-hoc linear function, denoted as $f_{\text{lin}}$, of $f$ at $\theta_0$:

$$f_{\text{lin}}(x; \theta_0 + \tau) = f(x; \theta_0) + \tau^\top \nabla_\theta f(x; \theta_0).$$

The obtained task vectors are denoted as $\mathcal{T}_{\text{lin}}$. The authors further demonstrated empirically that this approach yields an important property: $\mathcal{T}_{\text{lin}}$ is weight disentangled w.r.t. $f_{\text{lin}}$ at $\theta_0$.

To conclude this section, we define good task arithmetic performance as characterized by three key factors: high accuracy, computational efficiency, and good weight disentanglement.

## 3 Task Arithmetic in Attention Modules

### 3.1 Main Challenge of Task Arithmetic

Based on our previous definition, the performance of the unified model on each task $t$ can be simply decomposed as follows:

*Accuracy of unified models = Accuracy of individual models - Accuracy Gap.*

Unfortunately, existing studies reveal a trade-off between these factors. Non-linear fine-tuning can achieve high accuracy for individual models, but the resulting task vectors are less weight disentangled, leading to a larger accuracy gap in the unified model. Conversely, linear fine-tuning guarantees weight disentanglement and thus a lower accuracy gap, but the linearly fine-tuned individual models tend to be less accurate, as linear approximation brings error to the models. This trade-off presents a significant challenge in task arithmetic.

To this aim, we hypothesize that a combination of non-linear fine-tuning and weight disentanglement is necessary. This hypothesis motivates us to investigate the following question: Does there exist a sub-module within neural networks where non-linear fine-tuning can exhibit weight disentanglement? Our investigation reveals that the attention module is a promising candidate.

### 3.2 Accuracy Gap: Kernel Behavior and Weight Disentanglement of Attention Module

As per the study of NTK studies and the discussion in Ortiz-Jimenez et al. (2024), if a pretrained network $f(\cdot; \theta_0)$ exhibits *kernel behavior* during fine-tuning – that is, if the neural network updates primarily around its pretrained parameter initialization and can be approximated by its first-order Taylor expansion – then the resulting task vectors are weight disentangled. Directly examining whether kernel behavior can be challenging; however, it can be approximated by the following test.

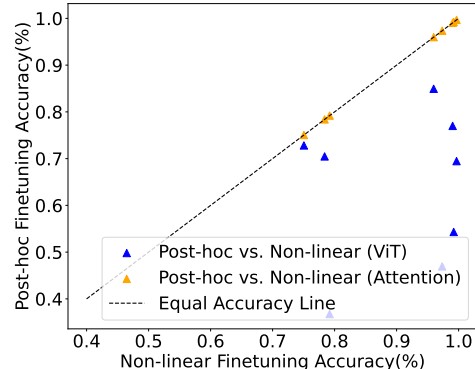

**Kernel Behavior Test.** Given a function $f$ with initial parameters $\theta_0$, and a task $t$ with task vector $\tau_t$, we define the Kernel Behavior Test as follows. If the equation

$$f(x; \theta_0 + \tau_t) = f_{\text{lin}}(x; \theta_0 + \tau_t),$$

Figure 3: Accuracy of non-linear and post-hoc models by tasks. The diagonal dashed line indicates post-hoc performance meets non-linear.

holds for all $x$ in the dataset of task $t$, we say that $f(\cdot; \theta)$ exhibits kernel behavior during fine-tuning using the given approach on task $t$, or more simply, that the given approach exhibits kernel behavior.

In our experiments, we compare the average accuracy across $T$ tasks of the non-linear function ($f(\cdot)$) and the post-hoc linear function ($f_{\text{lin}}$), fune-tuning on (1) all the parameters and (2) only on the attention modules. Specifically, we fine-tune several CLIP pre-trained ViTs (Dosovitskiy et al.,

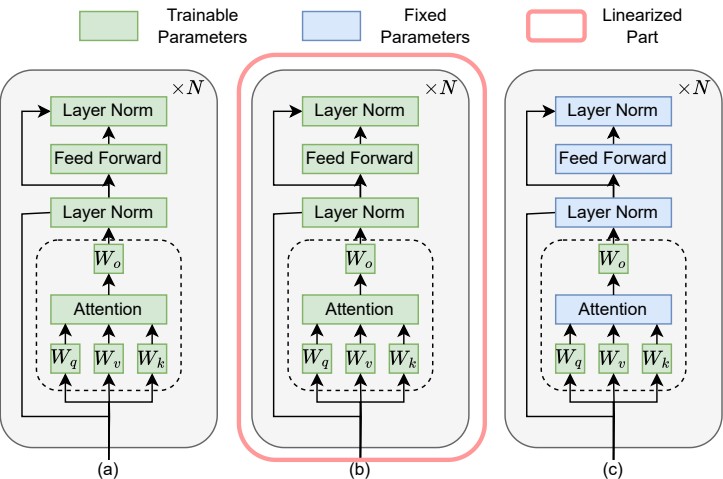

Figure 4: **Three types of fine-tuning paradigms.** (a) Non-linear fine-tuning where all the parameters will be updated. (b) Full-model linearization. (c) Attention modules only fine-tuning where only $W_q$, $W_v$, $W_k$ and $W_o$ will be updated. In this paper, we explore attention modules only fine-tuning.

2021) of different sizes following the same setup as Ilharco et al. (2023) on 8 tasks: Cars (Krause et al., 2013), DTD (Cimpoi et al., 2014), SUN397 (Xiao et al., 2016), EuroSAT (Helber et al., 2019), GTSRB (Stallkamp et al., 2011), MNIST (LeCun, 1998), SVHN (Netzer et al., 2011) and RESISC45 (Cheng et al., 2017).

The results in Figure 3 indicate that the attention module demonstrates kernel behavior. The triangle dots show the comparison of the kernel behavior test between the attention modules (yellow) and the whole models (blue), respectively. The proximity of dots to the diagonal dashed line indicates kernel behavior. The post-hoc of attention module, represented by yellow dots, consistently appears closer to the diagonal dashed line than the whole ViT (blue dots), suggesting superior performance.

This indicates that attention modules demonstrate stronger kernel behavior compared to the full model, suggesting that focusing on fine-tuning these modules could enhance the weight disentanglement, finally resulting in a low accuracy gap.

### 3.3 ACCURACY OF INDIVIDUAL MODELS WITH FINE-TUNING ATTENTION MODULES

Based on the kernel behavior of attention modules, we propose focusing on fine-tuning only the attention modules. The comparison of our fine-tuning paradigms with the non-linear fine-tuning paradigm and the NTK linearization fine-tuning paradigm is demonstrated in Figure 4. Next, we will show that fine-tuning attention modules also achieves high accuracy for individual models.

**Non-Linear Advantage.** We will first introduce a crucial concept referred to as *non-linear advantage*. For a given approach, the non-linear advantage is defined as the difference in accuracy of individual models between non-linear fine-tuning and the approach in question. Since non-linear fine-tuning typically achieves the highest accuracy for individual models, the non-linear advantage is always non-negative, i.e., non-linear advantage $\geq 0$.

**Accuracy of Individual Models.** In Figure 5, we demonstrate that fine-tuning attention modules can reduce the non-linear advantage—essentially improving the accuracy of individual models—compared to NTK linearization fine-tuning[2]. The figure presents two comparisons: 1) Round markers represent the comparison between our method and non-linear fine-tuning. 2) Triangular markers show the comparison between NTK linearization and non-linear fine-tuning.

The proximity of markers to the diagonal dashed line indicates a non-linear advantage equal to zero. Our method, represented by round markers, consistently appears closer to the diagonal dashed line than the NTK linearization (triangular markers), indicating a smaller non-linear advantage. This visual representation demonstrates that our approach of fine-tuning attention modules achieves per-

---

[2]Please see Appendix B for performance on each task.

Table 2: **Comparison of performance for task arithmetic across various visual models.** This table presents the average accuracy (%) and normalized accuracy (%) of various ViTs after incorporating the sum of task vectors from eight different tasks. We report results for the non-linear fine-tuning and NTK linearized models normalizing performance by their single-task accuracy.

| Method | ViT-B-32 | | ViT-B-16 | | ViT-L-14 | |
| --- | --- | --- | --- | --- | --- | --- |
| | Abs.(↑) | Norm.(↑) | Abs.(↑) | Norm.(↑) | Abs.(↑) | Norm.(↑) |
| Pre-trained | 48.40 | - | 55.25 | - | 66.40 | - |
| Non-linear Fine-tuning | 70.00 | 77.04 | 74.75 | 80.59 | 84.40 | 89.47 |
| NTK Linearization | 76.26 | 85.82 | 79.01 | 86.32 | 85.53 | 91.44 |
| **Ours** | **78.37** | **87.42** | **80.44** | **87.25** | **87.91** | **93.66** |

Table 3: **Comparison of performance for task arithmetic across different language models.** This table presents the average accuracy (%) of Flan-T5-base models on GLUE benchmark.

| | glue-cola | glue-mnli | glue-mrpc | glue-qqp | glue-rte | glue-sst2 | glue-stsb | avg |
| --- | --- | --- | --- | --- | --- | --- | --- | --- |
| Non-linear Fine-tuning | 79.87 | 80.94 | 59.31 | 82.19 | 50.54 | 89.33 | 70.55 | 73.25 |
| NTK Linearization | 75.93 | 83.19 | 76.72 | 87.88 | 62.09 | 92.09 | 66.51 | 77.77 |
| **Ours** | 80.63 | 86.25 | 86.76 | 89.69 | 72.20 | 93.92 | 87.67 | **85.30** |

formance closer to that of non-linear fine-tuning compared to NTK linearization, thus reducing the non-linear advantage more effectively.

## 3.4 ACCURACY OF UNIFIED MODELS WITH FINE-TUNING ATTENTION MODULES

We have demonstrated that our method has achieved both high accuracy for individual models and solved the accuracy gap. Then we will validate that our method has achieved high accuracy for unified models in terms of average accuracy and normalized accuracy.

To obtain the task vectors, we use the fine-tuned weights of different ViTs from before and use the same mixing coefficient for all tasks, i.e., $\alpha = \alpha_1 = \cdots = \alpha_T$ to ensure a fair comparison with Ortiz-Jimenez et al. (2024). We provide all the details of this experiment in Appendix A.

**Normalized Accuracy.** The normalized accuracy is calculated by the individual accuracy achieved by the model fine-tuned on each task. Mathematically,

$$\text{Normalized Acc} = \frac{1}{T} \sum_{t=1}^{T} \frac{[\text{acc}\,(f(x; \theta_0 + \sum_t \alpha_t \tau_t))]}{[\text{acc}\,(f(x; \theta_0 + \alpha_t \tau_t))]}.$$

**Accuracy of Unified Models.** We employ the benchmark proposed by Ilharco et al. (2023) to evaluate the task arithmetic ability of a pre-trained model, which consists of the 8 tasks described in Section 3.2. In particular, Table 2 shows that our method significantly outperforms its non-linear counterparts (Ilharco et al., 2023) and achieves state-of-the-art results on the task addition benchmarks. Our model achieves higher accuracy of the unified model through task addition (up to 2.38%). Additionally, our method not only outperforms on averaged accuracy but also on normalization accuracy.

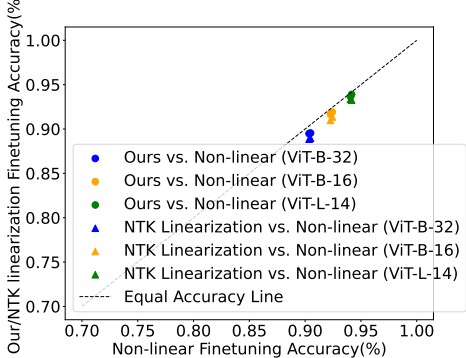

Figure 5: Averaged accuracy of non-linear and linear models. The diagonal dashed line indicates linear fine-tuning performance meets non-linear.

For Natual Language Processing (NLP) tasks, we utilize the Flan-T5 (Chung et al., 2022) as our pre-trained language model. For fine-tuning, we employ the Flan-T5-base models on seven tasks derived from the General Language Understanding Evaluation (GLUE) benchmark (Wang et al., 2018) with the same random seed 42 to initialize the models. These tasks are CoLA, MNLI, MRPC,

Table 4: **Efficiency comparison in terms of parameters, size, and training time.** The NTK linearization requires two to three times more computational resources and doubles the memory footprint compared to its non-linear counterpart. Our method outperforms NTK linearization in accuracy with only a quarter of the costs.

|  | Total Params | Trainable Params | Computational Cost (MB) | Training Time (Min) |
|---|---|---|---|---|
| Non-linear Fine-tuning | 222 M | 222 M | 891.614 | 17:38 |
| NTK Linearization | 470 M | 222 M | 1,881.926 | 40:59 |
| Ours | 222 M | 63.7 M | 891.614 | 10:15 |

QQP, RTE, SST2, and STSB. We report accuracy for all tasks and an average accuracy in Table 3, our method outperforms both non-linear fine-tuning and NTK linearization in vision and NLP tasks.

**Ablation Study.** To investigate whether we can enhance weight disentanglement performance by fine-tuning the Multiple-Layer Perceptron (MLP) modules in addition to attention modules, we conduct an ablation experiment with four paradigms: (1) fine-tuning only attention weights ($Q$, $K$, $V$, and $O$ projections) (ours), (2) fine-tuning attention weights and biases, (3) fine-tuning both attention and MLP weights, and (4) fine-tuning attention and MLP weights along with biases. Remarkably, all four paradigms outper-

Table 5: **Ablation study.** Single-task and performance of unified model (%) on 4 different settings.

| Paradigm | Single-task Accuracy(↑) | Multi-task Abs.(↑) | Multi-task Norm.(↑) |
|---|---|---|---|
| **(1)** | **89.55** | **78.37** | **87.42** |
| (2) | 89.48 | 77.71 | 86.79 |
| (3) | 88.95 | 76.52 | 86.11 |
| (4) | 89.43 | 77.80 | 86.93 |

formed NTK linearization in terms of both performance and weight disentanglement, indicating that ViT models exhibit strong kernel behavior within the attention modules and MLP. However, performance varied based on whether bias parameters were fine-tuned, with the best results aligning closely with settings used in LoRA. This suggests that further exploration of these configurations could yield valuable insights into optimizing task arithmetic.

**Efficiency Comparison.** Besides superior accuracy, our method is much more efficient than non-linear fine-tuning and NTK linearization as shown in Table 4 due to fine-tuning fewer parameters.

Our fine-tuning method significantly enhances the appeal of task arithmetic for practical applications. By improving the performance of individual models, our approach demonstrates the superiority of task arithmetic in achieving good accuracy of the unified model efficiently. Additionally, we have observed that kernel behavior within attention modules fosters greater task disentanglement. In the subsequent section, we will delve deeper into this concept, exploring its implications and potential for future advancements.

## 4 ROBUSTNESS OF TASK ARITHMETIC WITH RESPECT TO COEFFICIENT $\alpha$

In previous sections, we discuss how to find a good $\mathcal{T}$ in task arithmetic. Yet the robustness of task arithmetic (i.e., the effect of $\alpha$) on weight disentanglement has not been explored. To this end, in the following section, we first propose a metric to evaluate the weight disentanglement performance called "disentanglement error" on the representation module and prove that weight disentanglement emerges from the representation module. Then, we illustrate that our method has great weight disentanglement for a wider choice of $\alpha$ for both representation and classification modules, which demonstrates the robustness of the task arithmetic of our method.

### 4.1 WEIGHT DISENTANGLEMENT EMERGES FROM REPRESENTATION MODULE

To explore the robustness of task arithmetic and its effect on weight disentanglement, we propose a metric called "disentanglement error" to evaluate weight disentanglement performance. Unlike previous work that focused on task-specific modules, we investigate whether the representation module can satisfy Definition 2 (Weight Disentanglement) without relying on task-specific components.

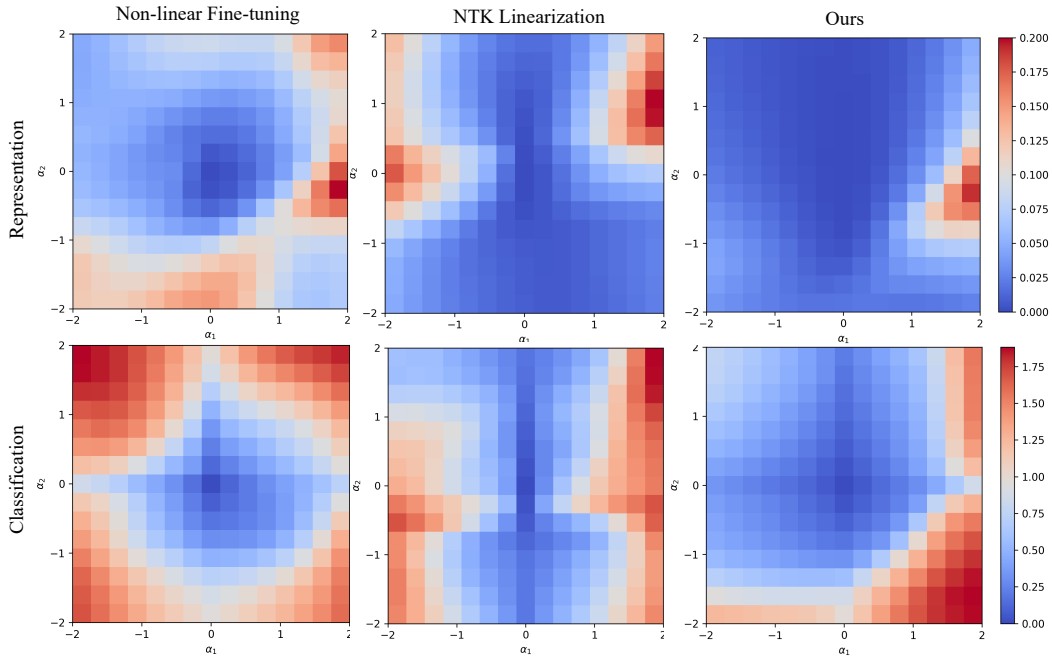

Figure 6: **Visualization of weight disentanglement.** The heatmaps show the disentanglement error $\xi(\alpha_1, \alpha_2)$ (see Eq. (2)) of a single representation module CLIP ViT-B/32 (top) and a combination of representation module and classification module on DTD - SUN397 task pair. We use prediction error for classification task as Ortiz-Jimenez did (Ortiz-Jimenez et al., 2024). Three fine-tuning paradigms are used from left to right: non-linear fine-tuning, NTK linearization, and ours. The blue regions denote areas of the weight space where weight disentanglement is stronger.

Our hypothesis is that pre-trained models can demonstrate task arithmetic properties independently of downstream tasks, maintaining consistent representations through task arithmetic. By focusing on the representation module alone, we aim to show that the inherent properties of pre-trained models are sufficient to support task arithmetic, potentially simplifying the process and broadening its applicability.

To visualize the level of weight disentanglement, we measure the discrepancy with Eq. (1) using the disentanglement error (Ortiz-Jimenez et al., 2024):

$$\xi(\alpha_1, \alpha_2) = \sum_{t=1}^{2} \mathbb{E}_{x \sim \mu_t} \left[ \text{dist} \left( f(x; \theta_0 + \alpha_t \tau_t), f(x; \theta_0 + \alpha_1 \tau_1 + \alpha_2 \tau_2) \right) \right], \quad (2)$$

where "dist" denotes any distance metric between output vectors. As we are dealing with representation distributions, in what follows we use the Kullback–Leibler divergence as the distance metric. In general, the smaller the value of $\xi(\alpha_1, \alpha_2)$ the more weight disentangled a model is at $(\alpha_1, \alpha_2)$.

### 4.2 WEIGHT DISENTANGLEMENT RESULTS

Figure 6 displays the disentanglement error of a CLIP ViT-B/32 model concerning several task vector pairs from different fine-tuning paradigms. We observe a minimal disentanglement error within a small region surrounding $\theta_0$, which enables task arithmetic. Different from disentanglement error at downstream tasks, it remains relatively small even for $\alpha_1, \alpha_2 > 1$, which indicates the power of task arithmetic has been limited by the performance of task-specific modules (classification head).

**Disentanglement Error Comparison.** Our method demonstrates greater weight disentanglement than its counterparts, as evidenced by the more extensive regions with low disentanglement errors in Figure 6 (right). This explains the higher normalized accuracy achieved (cf. Table 2) when fine-tuning attention modules only. The combination of greater weight disentanglement and better performance of individual models results in higher performance of the unified model.

These results demonstrate that our method achieves great weight disentanglement for a wider choice of $\alpha$ for both representation and classification modules, illustrating the robustness of task arithmetic in our approach.

## 5 RELATED WORK

Existing model merging techniques can be broadly categorized into two main types (Yang et al., 2024): (i) Pre-Merging Methods: These methods focus on enhancing the conditions necessary for effective model merging by optimizing the fine-tuning process of individual models. (ii) During Merging Methods: These approaches address task conflicts and interference through various strategies before executing the parameter merging operations.

**Pre-Merging Methods.** To establish better conditions for model merging, a significant body of work has focused on the fine-tuning processes of independent models. For instance, Ortiz-Jimenez et al. (2024) propose fine-tuning within the tangent space of the pre-trained model, leveraging the NTK framework to enhance performance during the fine-tuning stage. However, fine-tuning all parameters in the linearized model can be computationally intensive compared to nonlinear fine-tuning. To mitigate this issue, some studies advocate for selectively linearizing only certain layers; for example, Tang et al. (2023) suggest partially linearizing Adapter modules prior to merging them. Additionally, TAFT (Liu et al., 2024) introduces an efficient linearization method specifically for Transformer architectures, deriving closed-form linearized solutions that facilitate smoother integration of models. Overall, fine-tuning in the tangent space aids in disentangling both input and weight spaces, thereby reducing potential interference during subsequent model merging.

**During Merging Methods.** In the context of multi-task learning (MTL), model merging can be effectively achieved by employing various strategies to resolve task conflicts and perform parameter merging operations. Traditional methods often involve averaging or combining weights from multiple models to create a unified system, as demonstrated in prior works (Garipov et al., 2018; Ilharco et al., 2023; Wortsman et al., 2022). However, these basic merging techniques frequently underperform, particularly when tasks interfere with one another. Advanced methods have been developed to address this challenge by incorporating weighted-based strategies that assign different importance levels to task vectors during merging (Matena & Raffel, 2021; Ainsworth et al., 2023; Stoica et al., 2023; Yang et al., 2023). Furthermore, some approaches transform models into sparse subspaces before merging, effectively mitigating task interference and allowing for the removal of insignificant neurons from individual models while enabling the combination of multiple sparse models within a parameter subspace (Yadav et al., 2023; Tam et al., 2023; Li et al., 2023; Zhang et al., 2023; Huh et al., 2024; Huang et al., 2024). This innovative perspective opens new avenues for model merging, enhancing overall performance and flexibility in multi-task applications.

Our method falls into the Pre-Merging category, focusing on the fine-tuning process and achieving superior performance in task arithmetic with high efficiency.

## 6 DISCUSSION

In this work, we conducted a comprehensive analysis of task arithmetic in deep neural networks, uncovering its fundamental mechanisms and enhancing its performance. Our findings reveal that attention modules exhibit kernel behavior, leading to improved weight disentanglement when fine-tuned exclusively, without compromising individual accuracy or efficiency. Crucially, we demonstrated that weight disentanglement emerges primarily from the representation module, while task-specific modules can limit the effectiveness of task arithmetic. This insight opens up new possibilities for applying task arithmetic in unsupervised learning scenarios and broadens its potential applications.

While our approach significantly advances the field of task arithmetic, several limitations and opportunities for future research remain. Current task vectors are constrained to models with identical architectures and initializations due to their reliance on element-wise weight operations. Future studies could explore integrating task arithmetic with partial fine-tuning techniques, focusing on varying numbers of attention blocks. Additionally, investigating the relationship between the sparsity of attention modules and their kernel behavior may provide insights into learnable tasks. Understanding the nuanced impact of fine-tuning bias on model performance and weight disentanglement also presents an important avenue for future research. These investigations could lead to more robust and efficient methods for adapting pre-trained models to various tasks, significantly enhancing their applicability and effectiveness in real-world scenarios.

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

# A EXPERIMENTAL DETAILS

All our experiments are performed using the same hardware consisting of four 3090 NVIDIA GPUs with 24GB of memory each, which can be reproduced in less than 150 GPU hours. The details of each experiment are the following.

**Datasets.** We evaluate task arithmetic on a set of popular benchmark datasets from various domains. The dataset collection includes:

- **SVHN** (Netzer et al., 2011): The Street View House Numbers dataset is a real-world image dataset for developing machine learning and object recognition algorithms with minimal requirements on data preprocessing and formatting.
- **MNIST** (LeCun, 1998): A database of handwritten digits, with 60,000 training images and 10,000 testing images.
- **EuroSAT** (Helber et al., 2019): A dataset based on Sentinel-2 satellite images covering 13 spectral bands, with 10 classes and a total of 27,000 labeled and geo-referenced images.
- **RESISC45** (Cheng et al., 2017): The remote sensing image scene classification dataset, consisting of 31,500 images in 45 scene classes.
- **Cars** (Krause et al., 2013): This dataset contains images of cars categorized into various fine-grained classes. It is widely used for fine-grained image classification tasks, providing a rich set of vehicle images for training and evaluation.
- **DTD (Describable Textures Dataset)** (Cimpoi et al., 2014): This dataset is designed for texture recognition and categorization. It consists of texture images organized into 47 categories, each labeled with attributes describing the texture patterns. It is commonly used to evaluate texture recognition algorithms.
- **SUN397** (Xiao et al., 2016): The Scene UNderstanding (SUN) dataset is a large-scale dataset for scene recognition, containing 397 categories with a total of over 100,000 images. It is used to evaluate scene understanding models and to benchmark scene classification algorithms.
- **GTSRB (German Traffic Sign Recognition Benchmark)** (Stallkamp et al., 2011): This dataset comprises images of German traffic signs, classified into over 40 categories. It is used to develop and evaluate traffic sign recognition systems, particularly in the context of autonomous driving and intelligent transportation systems.

**Fine-tuning.** All the fine-tuning experiments follow the same training protocol specified in Ilharco et al. (Ilharco et al., 2022) with minor modifications to the training code to use linearized models when needed. In particular, we fine-tune all datasets starting from the same CLIP pre-trained checkpoint downloaded from the open_clip repository (Cherti et al., 2023). We fine-tune for 2,000 iterations with a batch size of 128, a learning rate of $10^{-5}$ and a cosine annealing learning rate schedule with 200 warm-up steps and the AdamW optimizer (Loshchilov & Hutter, 2019). As introduced in Ilharco et al. (Ilharco et al., 2022), during fine-tuning, we freeze the weights of the classification layer obtained by encoding a standard set of zero-shot template prompts for each dataset. Freezing this layer does not harm accuracy and ensures that no additional learnable parameters are introduced during fine-tuning (Ilharco et al., 2022). We use this exact same protocol to fine-tune the non-linear and linearized models.

**Tuning of $\alpha$ in Task Arithmetic Benchmarks.** As in Ilharco et al. (Ilharco et al., 2022), we use a single coefficient $\alpha$ to tune the size of the task vectors used to modify the pre-trained models. This is equivalent to setting $\alpha = \alpha_1 = \ldots = \alpha_T$ in Eq. 1. In the task addition benchmarks, after fine-tuning, we evaluate different scaling coefficients $\alpha \in \{0.0, 0.05, 0.1, \ldots, 1.0\}$ and choose the value that achieves the highest target metric on a small held-out proportion of the training set as specified in Ilharco et al. (Ilharco et al., 2022). Namely, maximum normalized average accuracy, and minimum target accuracy on each dataset that still retains at least 95% of the accuracy of the pre-trained model on the control task. The tuning of $\alpha$ is done independently for non-linear fine-tuning, linearized fine-tuning, and post-hoc linearization.

**Disentanglement Error.** To produce the weight disentanglement visualizations of Figure 6, we compute the value of $\xi(\alpha_1, \alpha_2)$ on a $15 \times 15$ grid of equispaced values in $[-2, 2] \times [-2, 2]$. To estimate the disentanglement error, we use a random subset of 2,048 test points for each dataset.

## B    FURTHER EXPERIMENTAL RESULTS

We now present additional experiments that expand the findings discussed in the main text.

**Fine-tuning Accuracy.**    In Figure 7, we report the single-task accuracy achieved by different CLIP models after fine-tuning with different approaches (referred to as non-linear, NTK linearization, and our method).

**Weight Disentanglement on Different Task Pairs.**    In Figure 8, we illustrate weight disentanglement on different task pairs.

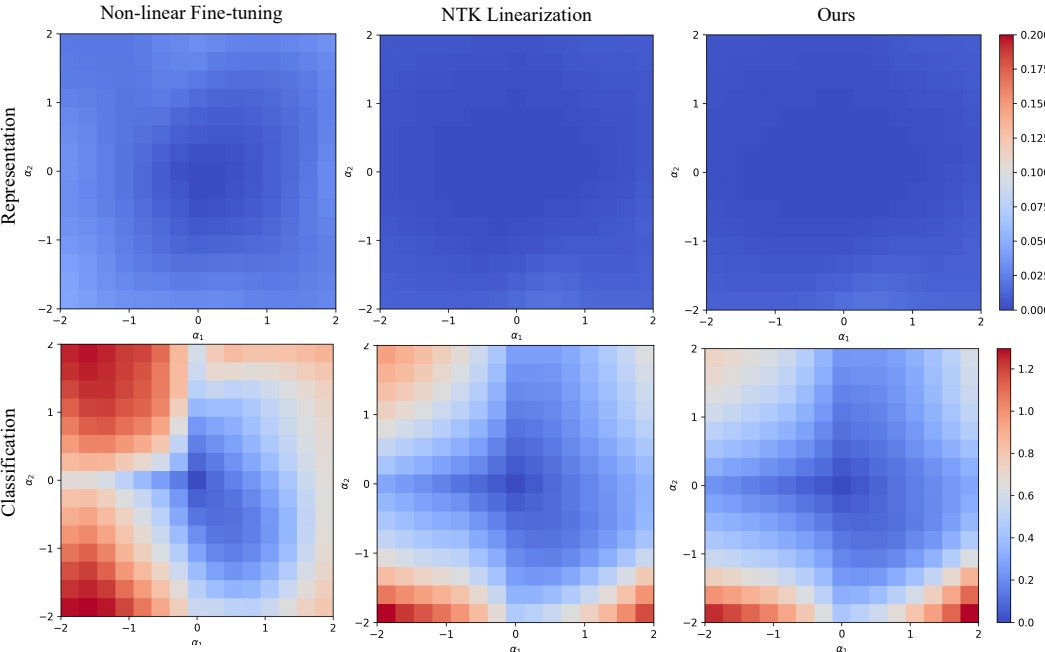

Figure 8: **Visualization of weight disentanglement.** The heatmaps show the disentanglement error $\xi(\alpha_1, \alpha_2)$ of a single representation module of CLIP ViT-B/32 (top) and a combination of representation module and classification module (bottom) on Cars - RESISC45 task pair. Three fine-tuning paradigms are used from left to right: non-linear fine-tuning, NTK linearization, and ours. The light regions denote areas of the weight space where weight disentanglement is stronger.

**Parameter Sensitivity Analysis.**    In our experiments, we replicate the experimental setup used by Ortiz-Jimenez et al. (2024) to evaluate the impact of varying $\alpha$ coefficients on model performance. The results are summarized in Figure 9, which demonstrates that our method exhibits greater robustness across different choices of $\alpha$ compared to both non-linear fine-tuning and NTK linearization. As illustrated, our method consistently outperforms both non-linear fine-tuning and NTK linearization across a wide range of $\alpha$ values, indicating its robustness in maintaining performance even with varying coefficients.

**Similarity between Task Vectors.**    Figure 10 shows the cosine similarity between task vectors from ViT for three types of fine-tuning (cf. Figure 4) on image classification tasks. Vectors from attention modules only fine-tuning are closer to orthogonal than those from both non-linear fine-tuning and NTK linearization, indicating that models fine-tuned with full parameters are more independent. This finding aligns with discussions in (Ilharco et al., 2023; Tang et al., 2023) and is supported by the experimental results in Table 2. The experimental details are described in Appendix A.

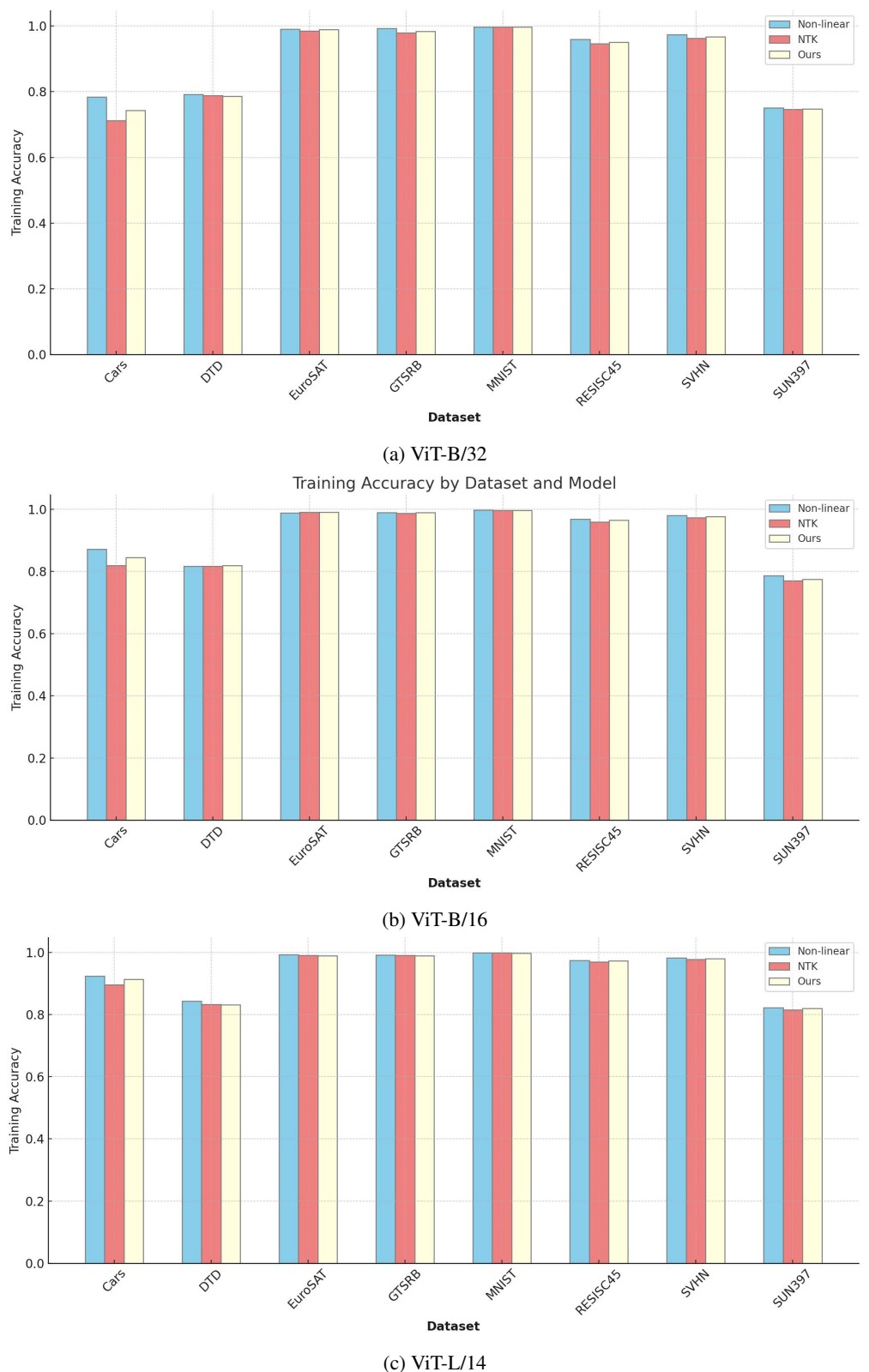

(a) ViT-B/32

(b) ViT-B/16

(c) ViT-L/14

Figure 7: Single-task accuracy of different models obtained using different strategies on each of the tasks.

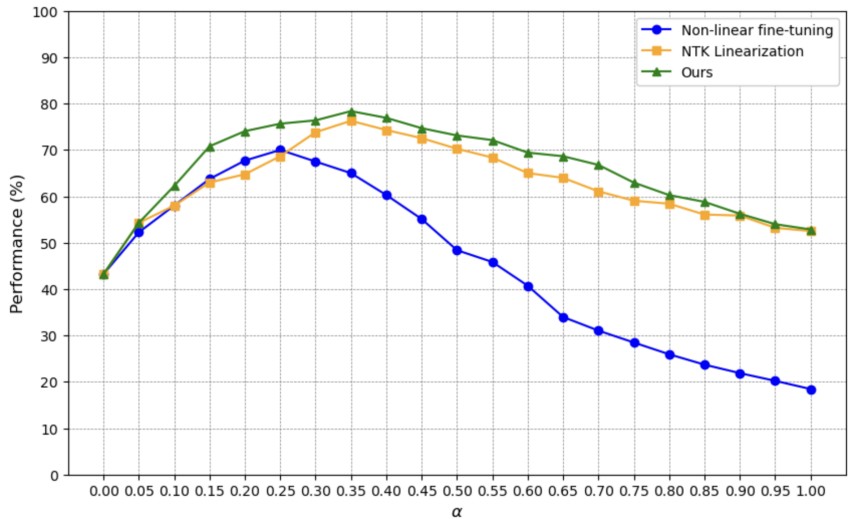

Figure 9: Performance comparison across different methods with varying $\alpha$ values.

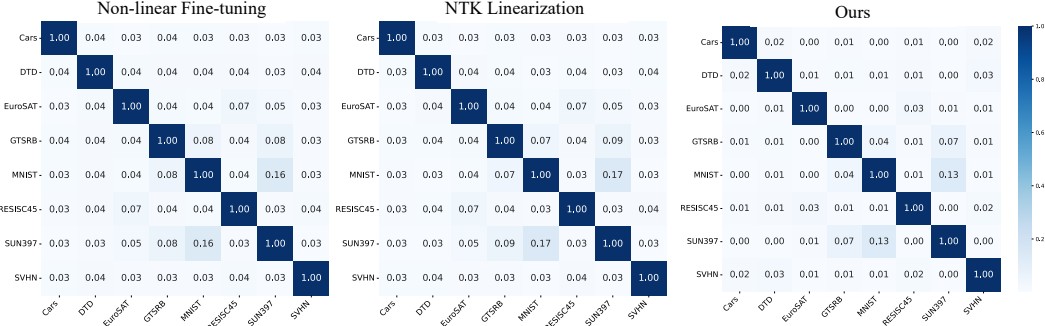

Figure 10: **Similarity heatmaps.** These figures show heatmaps of the cosine similarity between task vectors from task-specific CLIP models (Radford et al., 2021) fine-tuned on different tasks. Three fine-tuning paradigms from left to right: non-linear fine-tuning, NTK linearization, and Attention modules only fine-tuning (Ours).

## C  IMPACT STATEMENT

This paper presents work whose goal is to advance the field of Machine Learning. There are many potential societal consequences of our work, none of which we feel must be specifically highlighted here.

