# OpenReview forum: "Fine-Tuning Attention Modules Only: Enhancing Weight Disentanglement in Task Arithmetic"
_ICLR.cc/2025/Conference — ICLR 2025 Poster_

### Official Review · Reviewer_Rsur · 2024-10-27

**Soundness:** 3
**Presentation:** 2
**Contribution:** 3
**Rating:** 8
**Confidence:** 5

**Summary:**

The authors propose to improve the performance of weight-interpolation-based model merging methods such as Task Arithmetic by finding a sub-module within the deep neural networks. I strongly agree with this idea because, for existing work based on NTK linearization, the model's performance on individual downstream tasks has been compromised in order to achieve better performance in multi-task model merging. This trade-off is usually unacceptable in practical applications, where ensuring performance on downstream tasks is paramount for fine-tuning. However, there are some typos and minor mistakes in this manuscript, which need to be addressed to improve the overall clarity and professionalism of the document.

Therefore, I would like to assign this manuscript an overall score of 6, indicating that it requires minor revisions. However, **I am open to raising the score if the authors address the specific concerns and suggestions outlined in my detailed review.**

**Strengths:**

1. The motivation of this study is clear. The authors seek to identify a subset of parameters that can undergo non-linear fine-tuning while retaining linear characteristics, such as those observed in NTK linearization. Consequently, the attention modules, particularly the linear layers within them, are selected for this purpose.
2. The experimental results presented in Figure 3 demonstrate the validity of the proposed method.
3. Models of varying sizes (ViT-B-32, B-16 and L-14) are employed for experimental evaluation.
4. The authors conduct experiments on both the vision domain and NLP tasks.

**Weaknesses:**

1. The task arithmetic presented in Definition 1 (lines 167-175) deviates from standard practice. In the original paper [1], task arithmetic is defined as $\theta_{unified} = \theta_0 + \lambda \sum_{i=1}^T \tau_i$, employing only a single scaling factor. Otherwise, it should be explicitly stated as task-wise AdaMerging [2].
2. Missing scaling factor of task vectors in lines 52-53 and lines 353-357.
3. Is the summation symbol missing on the right-hand side of Eq.(1) (lines 196-199)?
$$ RHS = \sum_{t=1}^T \dots $$
4. In lines 400-401, the "disentanglement error" was first proposed by Ortiz-Jimenez et al. (2024).
5. Recent work on deep model fusion is missing from the related literature section. The authors may refer to these surveys [3,4,5] for additional information.
6. A minor suggestion: The results for NLP tasks (Table 4) should be included in the main text.
7. The ablation study in Appendix B.4 is not mentioned in the main text. Consider including it in the main text.

[1] Ilharco et al. Editing models with task arithmetic. In International Conference on Learning Representations. ICLR 2023.
[2] Yang et al. AdaMerging: Adaptive Model Merging for Multi-Task Learning. ICLR 2024.
[3] Tang et al. FusionBench: A Comprehensive Benchmark of Deep Model Fusion.
[4] Yadav et al. A Survey on Model MoErging: Recycling and Routing Among Specialized Experts for Collaborative Learning.
[5] Yang et al. Model Merging in LLMs, MLLMs, and Beyond: Methods, Theories, Applications and Opportunities.

**Questions:**

1. The differences between Definition 2 in the manuscript and the original definition in Ortiz-Jimenez et al. (2024) remain unclear. Could the authors elaborate further on these differences?

---

> ### Author Response · Authors · 2024-11-21
> **Rebuttal**
>
> Thank you for your positive feedback and thoughtful evaluation of our manuscript. We greatly appreciate your recognition of the motivation behind our study and the importance of addressing the trade-offs associated with existing model merging methods, particularly those based on NTK linearization.
>
> We notice that there are some typos in the manuscript, and we are committed to fixing them to enhance the overall clarity and professionalism of our document. Your detailed review will be invaluable in guiding us through these revisions.
>
> **Weaknesses:**
>
> > 1.The task arithmetic presented in Definition 1 (lines 167-175) deviates from standard practice. In the original paper [1], task arithmetic is defined as $θ_{unified}=θ_0+λ∑_{i=1}^Tτ_i$, employing only a single scaling factor. Otherwise, it should be explicitly stated as task-wise AdaMerging [2].
> >
>
> Response:
>
> We appreciate this observation about Definition 1. While [1] introduced task arithmetic conceptually, it did not provide an explicit mathematical definition. Our formulation in Definition 1 presents a generalized form that aligns with Eq. (1) in [6], allowing for task-specific scaling factors (α_t).
> In our implementation (line 348), we use identical mixing coefficients across all tasks (α = α_1 = ... = α_T) to ensure fair comparison with prior work [6]. This practical choice maintains consistency with established practices while our definition preserves flexibility for future extensions, including task-wise approaches like AdaMerging [2].
>
> Here is the revision in our paper: modify line 348 in the original paper to include [6] to indicate the implementation on equal $\alpha$ is a fair comparison.
>
> > 2.Missing scaling factor of task vectors in lines 52-53 and lines 353-357.
> >
>
> Response:
>
> Thank you for highlighting the omission of the scaling factor of task vectors in lines 52-53 and lines 353-357. We appreciate your attention to detail. Here is our rebuttal:
>
> We acknowledge the oversight regarding the missing scaling factor ($\alpha$) in our initial presentation. In the introduction, we aimed to convey our conceptual progression from an idealized view to a more practical application, which led to the omission of $\alpha$ for simplicity. However, we recognize that this may have caused confusion.
>
> To clarify and enhance the specificity of our definitions, we have now added the scaling factor $\alpha$ to both equations in the revised manuscript. This adjustment ensures that our definitions accurately reflect the standard formulation used in task arithmetic, maintaining consistency with existing literature while also improving clarity for readers.
>
> We appreciate your feedback, which has helped us improve the precision of our work. Thank you for your understanding as we strive to present our ideas clearly and effectively.
>
> > 3.Is the summation symbol missing on the right-hand side of Eq.(1) (lines 196-199)?
> >
> >
> > $RHS=∑_{t=1}^T…$
> >
>
> Response:
>
> Thank you for your careful observation regarding the missing summation symbol on the right-hand side of Eq. (1). We appreciate the opportunity to correct this oversight.
>
> We have reviewed the equation and agree that the summation symbol should indeed be included to accurately reflect the intended mathematical formulation. We have added the summation symbol to the right-hand side of Eq. (1) as per previous work, ensuring clarity and consistency in our presentation.
>
> This correction will enhance the precision of our definitions and align our notation with established conventions in the field. We appreciate your feedback, which helps us improve the quality and accuracy of our manuscript. Thank you for your understanding.
>
> > 4.In lines 400-401, the "disentanglement error" was first proposed by Ortiz-Jimenez et al. (2024).
> >
>
> Response:
>
> Thank you for bringing this to our attention. We apologize for the oversight in not properly attributing the concept of "disentanglement error." You are correct that this concept was first introduced by Ortiz-Jimenez et al. (2024). We would like to offer the following clarification and correction:
> Our definition of "disentanglement error" in lines 400-401 is indeed based on the work of Ortiz-Jimenez et al. (2024). We should have explicitly cited their paper when introducing this concept. The omission was unintentional, and we appreciate the opportunity to correct this.
>
> Our use of this metric is consistent with its original formulation, and it plays a crucial role in our analysis of weight disentanglement. We believe that properly attributing this concept strengthens our paper by acknowledging the important prior work in this field.
> We thank the reviewer for their careful attention to detail, which has allowed us to correct this oversight and improve the accuracy of our citations. We are committed to ensuring proper attribution of all concepts and methods used in our research.

---

> ### Author Response · Authors · 2024-11-21
> **Rebuttal (Part 2)**
>
> > 5.Recent work on deep model fusion is missing from the related literature section. The authors may refer to these surveys [3,4,5] for additional information.
> >
>
> Response:
> Thanks for the valuable suggestion, we have updated Related Work section, please refer to general response.
>
> > 6.A minor suggestion: The results for NLP tasks (Table 4) should be included in the main text.
> >
>
> Response:
>
> Thanks for the suggestion, we have included NLP results in the main text.
>
> > 7.The ablation study in Appendix B.4 is not mentioned in the main text. Consider including it in the main text.
> >
>
> Response:
> Thanks for the suggestion, we have included ablation study in the main text.
>
> [1] Ilharco et al. Editing models with task arithmetic. In International Conference on Learning Representations. ICLR 2023. [2] Yang et al. AdaMerging: Adaptive Model Merging for Multi-Task Learning. ICLR 2024. [3] Tang et al. FusionBench: A Comprehensive Benchmark of Deep Model Fusion. [4] Yadav et al. A Survey on Model MoErging: Recycling and Routing Among Specialized Experts for Collaborative Learning. [5] Yang et al. Model Merging in LLMs, MLLMs, and Beyond: Methods, Theories, Applications and Opportunities.
>
> [6] Task arithmetic in the tangent space: Improved editing of pre-trained models (NeurlPS 2024)
>
> **Questions:**
>
> > 1. The differences between Definition 2 in the manuscript and the original definition in Ortiz-Jimenez et al. (2024) remain unclear. Could the authors elaborate further on these differences?
> >
>
> Response:
>
> Thank you for your insightful question regarding the differences between Definition 2 in our manuscript and the original definition presented in Ortiz-Jimenez et al. (2024). We appreciate the opportunity to elaborate on these distinctions, especially as we made significant efforts to maintain a similar formulation in our definition to ensure consistency with prior work.
>
> 1. **Characterization of Weight Disentanglement**:
> Our definition characterizes weight disentanglement as a property of a set of task vectors with respect to the function $f$, whereas the original definition characterizes it as a property of the function $f$ with respect to the task vectors. This intentional reversal is significant for two reasons:
>     - It aligns more closely with the term "weight disentanglement," which inherently refers to the disentanglement of different weights (task vectors) rather than the function itself.
>     - Both our work and Ortiz-Jimenez et al. (2024) aim to find better task vectors for task arithmetic, rather than improving the function $f$.
> 2. **Focus on the Representation Module**:
> Our definition specifically applies to the representation module, whereas the original definition encompasses the entire neural network. This narrower focus is motivated by our observation that task arithmetic is performed exclusively on the representation module. We demonstrate in Section 4 that this focus is well-justified, as weight disentanglement primarily emerges from this module.
> 3. **Mathematical Formulation**:
> While the core concept remains similar, our mathematical formulation differs slightly in reflecting these conceptual changes. This allows for a more precise analysis of weight disentanglement in the context of the representation module.
>
>     We believe these differences, although subtle, are crucial for advancing the understanding and application of weight disentanglement in task arithmetic. Our intention was not to contradict the original definition but to refine it for our specific context and goals while maintaining a consistent formulation.
>
>
> Thank you once again for your valuable feedback, which helps us improve our manuscript and clarify our contributions to this important area of research.

---

> ### Comment · Reviewer_Rsur · 2024-11-22
>
> After reading comments from other reviewers and the authors' responses, most of my concerns have been addressed. This study proposes a simple and effective pre-merging method for improved task arithmetic.
>
> Minor concerns:
> 1. The scaling factor $\alpha$ is missing from the equation in Figure 1 of the unified model.
> 2. It is important to conduct the parameter sensitivity analysis to evaluate the impact of varying $\alpha$ as pointed out by reviewer CQUv. Just personal preference: Table 6 in the revised manuscript is not very intuitive, I would prefer to draw an *additional* line plot.

---

> > ### Author Response · Authors · 2024-11-22
> > **Reply by Authors**
> >
> > Thank you for your further suggestions, we have updated the manuscript accordingly and hope all your concerns are addressed! If any additional details are required, please feel free to ask!

---

> ### Comment · Reviewer_Rsur · 2024-11-22
>
> All my concerns have been addressed. I am raising my rating to accept.

---

> > ### Author Response · Authors · 2024-11-22
> > **Thank You!**
> >
> > Thank you very much for raising your score. We really appreciate all your comments and suggestions.

---

### Official Review · Reviewer_6SAc · 2024-10-31

**Soundness:** 3
**Presentation:** 4
**Contribution:** 3
**Rating:** 6
**Confidence:** 3

**Summary:**

This paper proposes a new method to enhance weight decoupling in task arithmetic by fine-tuning only the attention module in the Transformer model. This method improves the weight decoupling ability of the unified model while maintaining the performance of individual models, thereby improving the performance of task arithmetic.

**Strengths:**

Innovation: The proposed method of fine-tuning only the attention module is innovative in the field of task arithmetic and can effectively solve the weight decoupling problem.
Experimental design: The experimental part covers multiple data sets and tasks, enabling a comprehensive evaluation of the effectiveness of the proposed method.
The results are remarkable: the method proposed in the paper achieves better performance than existing techniques on multiple benchmarks.

**Weaknesses:**

Parameter sensitivity analysis: The paper mentions the effect of the choice of α coefficient on performance, but does not provide a detailed sensitivity analysis. It is recommended to add experiments on the effect of the choice of α coefficient on model performance.
Computational resource consumption: The paper mentions the efficiency advantage of the proposed method, but does not provide a direct comparison with existing methods in terms of computational resource consumption. It is recommended to add analysis in this regard, especially memory and time consumption in actual deployment.

**Questions:**

While the paper mentions some related work, the relationship to existing methods could be discussed in more detail, especially the latest advances in model fusion and weight interpolation.

---

> ### Author Response · Authors · 2024-11-21
> **Rebuttal**
>
> **Weaknesses:**
>
> > 1. Parameter sensitivity analysis: The paper mentions the effect of the choice of α coefficient on performance, but does not provide a detailed sensitivity analysis. It is recommended to add experiments on the effect of the choice of α coefficient on model performance.
> >
>
> Response:
>
> Thank you for your valuable suggestion regarding the parameter sensitivity analysis of the α coefficient. We appreciate the opportunity to address this concern and provide additional insights into our findings.
> In our experiments, we replicated the experimental setup used by Ortiz-Jimenez et al. (2024) to evaluate the impact of varying α coefficients on model performance. The results are summarized in the table below, which demonstrates that our method exhibits greater robustness across different choices of α compared to both non-linear fine-tuning and NTK linearization:
>
> | $\alpha$ | Non-linear fine-tuning | NTK Linearization | ours |
> | --- | --- | --- | --- |
> | 0 | 43.21% | 43.21% | 43.21% |
> | 0.05 | 52.24% | 54.28% | 54.28% |
> | 0.1 | 58.00% | 57.99% | 62.23% |
> | 0.15 | 63.74% | 62.98% | 70.79% |
> | 0.2 | 67.70% | 64.75% | 74.05% |
> | 0.25 | **70.00%** | 68.65% | 75.66% |
> | 0.3 | 67.49% | 73.82% | 76.38% |
> | 0.35 | 65.00% | **76.28%** | **78.37%** |
> | 0.4 | 60.28% | 74.29% | 76.92% |
> | 0.45 | 55.10% | 72.54% | 74.69% |
> | 0.5 | 48.38% | 70.25% | 73.12% |
> | 0.55 | 45.84% | 68.36% | 72.12% |
> | 0.6 | 40.72% | 65.01% | 69.42% |
> | 0.65 | 33.98% | 63.98% | 68.64% |
> | 0.7 | 31.04% | 61.07% | 66.77% |
> | 0.75 | 28.48% | 59.05% | 62.96% |
> | 0.8 | 25.94% | 58.40% | 60.26% |
> | 0.85 | 23.70% | 56.07% | 58.80% |
> | 0.9 | 21.87% | 55.85% | 56.23% |
> | 0.95 | 20.21% | 53.20% | 53.99% |
> | 1 | 18.44% | 52.51% | 52.85% |
>
> As illustrated, our method consistently outperforms both non-linear fine-tuning and NTK linearization across a wide range of α values, indicating its robustness in maintaining performance even with varying coefficients. We have included this experiment in the appendix as Figure 9.
>
> > 2.Computational resource consumption: The paper mentions the efficiency advantage of the proposed method, but does not provide a direct comparison with existing methods in terms of computational resource consumption. It is recommended to add analysis in this regard, especially memory and time consumption in actual deployment.
> >
>
> Response:
>
> We appreciate the reviewer's attention to the computational efficiency of our method. We would like to highlight that Table 3 in our paper directly addresses this concern by providing a comprehensive comparison of computational resource consumption between our method and existing approaches. Specifically:
>
> 1. Parameters: Our method uses 222M total parameters, with 63.7M trainable parameters, which is significantly fewer than the 470M total parameters required by NTK linearization.
> 2. Model Size: Our approach maintains the same estimated size (891.614 MB) as non-linear fine-tuning, while NTK linearization more than doubles the memory footprint to 1,881.926 MB.
> 3. Training Time: Our method achieves the fastest training time at 10:15 minutes, compared to 17:38 minutes for non-linear fine-tuning and 40:59 minutes for NTK linearization.
>
> We also put the Table 3 in here for your convenience to review.
>
> |  | Total Params | Trainable Params | Computational Cost (MB) | Training Time (Min) |
> | --- | --- | --- | --- | --- |
> | Non-linear Fine-tuning | 222 M | 222 M | 891.614 | 17:38 |
> | NTK Linearization | 470 M | 222 M | 1,881.926 | 40:59 |
> | Ours | 222 M | 63.7 M | 891.614 | 10:15 |
>
> These metrics clearly demonstrate that our approach not only improves performance but also significantly reduces computational resource consumption compared to existing methods, particularly NTK linearization. Our method achieves superior accuracy while using only about a quarter of the computational resources required by NTK linearization.
>
> We believe this efficiency, combined with our improved performance, makes our method particularly suitable for real-world applications and large-scale deployments. The reduced resource requirements also make our approach more accessible to researchers and practitioners with limited computational resources.
>
> We thank the reviewer for highlighting this important aspect, and we hope this clarification adequately addresses the concern about computational resource consumption.
>
> **Questions:**
>
> > While the paper mentions some related work, the relationship to existing methods could be discussed in more detail, especially the latest advances in model fusion and weight interpolation.
> >
>
> Response:
>
> Thanks for the valuable suggestion, we have updated Related Work section, please refer to general response.

---

### Official Review · Reviewer_CQUv · 2024-11-02

**Soundness:** 2
**Presentation:** 3
**Contribution:** 1
**Rating:** 3
**Confidence:** 4

**Summary:**

The paper addresses task interference issues in task arithmetic, which arise when combining weights from multiple tasks into a single unified model. To tackle this issue, it uses Neural Tangent Kernel (NTK) linearization and trains only the attention layers in the transformer to reduce training costs. Through several experiments, the paper demonstrates the efficiency of the proposed approach and explains task arithmetic by distinguishing the roles of the representation module and the task-specific module.

**Strengths:**

1. The paper is easy to follow and provides clear explanations of essential concepts, making it easier to understand both the proposed approach and the fundamentals of task arithmetic.

**Weaknesses:**

1. The paper appears to be a naive combination of prior work, specifically [1] and [2], with modifications focused on training parameter selection (e.g., training only the attention layers in transformers), which limits its novelty.

[1] Multitask prompted training enables zero-shot task generalization (ICLR 2023)
[2] Task arithmetic in the tangent space: Improved editing of pre-trained models (NeurlPS 2024)

2. The main contribution claimed by the authors—asserting that weight disentanglement primarily arises from the representation module while the effectiveness of task arithmetic is constrained by task-specific components—is not well-supported. Additional ablation studies, analysis, or theoretical support are needed to substantiate this claim. Depending on interpretation, the proposed contribution may be a trivial insight already known in the fields of task arithmetic or multi-task learning.

3. The experiments are quite limited, as many related works cited by the authors are missing from the experimental results, particularly in Tables 2 and 3. This omission reduces the validity of the proposed method.

4. The paper's theoretical contributions rely heavily on [2], which limits its originality and overall contribution.

**Questions:**

Refer to the weaknesses section.

---

> ### Author Response · Authors · 2024-11-21
> **Rebuttal**
>
> > 1. The paper appears to be a naive combination of prior work, specifically [1] and [2], with modifications focused on training parameter selection (e.g., training only the attention layers in transformers), which limits its novelty.
>
> Response:
>
> We appreciate the reviewers' thoughtful comments but we respectfully point out your assertion that our paper is a naive combination of [1,2] is not true, as our work is not related to [1] which is multitask prompted training and not incremental from [2] which is ntk linearization to enforce model into kernel behavior. Here are the details:
>
> 1. Our work is fundamentally different from [1] (ICLR 2023).
> While [1] focuses on zero-shot task generalization through multitask prompted training, our paper addresses the challenge of task arithmetic - combining multiple fine-tuned models efficiently. These are distinct problems with different goals and methodologies.
> 2. Our work is not an incremental work from [2] (NeurIPS 2024)
> Unlike [2] (NeurIPS 2024), which applies NTK linearization to the entire model, we discovered that kernel behavior naturally emerges within attention modules. This is a novel finding that has not been explored in previous work on task arithmetic.
> 3. We propose fine-tuning only the attention modules (Wq, Wv, Wk, Wo), which is distinctly different from [2]'s approach of linearizing the entire model. This targeted approach significantly improves efficiency while maintaining performance.
>
> In summary, while our work builds upon existing research in the field, it presents significant novel contributions in terms of methodology, analysis, and performance improvements. Our approach offers a new perspective on task arithmetic that is distinct from and complementary to prior work, rather than being a naive combination of existing methods.
>
> > 2.1 The main contribution claimed by the authors—asserting that weight disentanglement primarily arises from the representation module while the effectiveness of task arithmetic is constrained by task-specific components—is not well-supported. Additional ablation studies, analysis, or theoretical support are needed to substantiate this claim.
> >
>
> Response:
>
> Thank you for your comment and suggestion. We would like to clarify that in addition to the ablation studies in the main paper, we have conducted additional ablation studies in Appendix B.4 to support our claim.
>
> First, in the main paper, we provide substantial empirical evidence to support our claim through our disentanglement error analysis (Section 4). The ablation studies in Figure 6 and Figure 8 demonstrate disentanglement errors of both representation module and combination model, where the representation module exhibits strong weight disentanglement properties, even for coefficient values α1, α2 > 1. This is in contrast to the behavior observed when considering the entire model including task-specific components.
>
> In addition to the direct ablation comparison of with and without the task-specific components in the experiments shown in Figure 6, we have conducted the following additional ablation studies to support our claim. Specifically, they include four paradigms: (1) only fine-tuning attention weights (Q, K, V, and O projections) (ours),
>
> (2) fine-tuning attention weights and bias,
>
> (3) fine-tuning attention and MLP weights,
>
> (4) fine-tuning attention and MLP weights and bias.
>
> The results further verify that weight disentanglement arising primarily from the representation module.
>
> > 2.2 Depending on interpretation, the proposed contribution may be a trivial insight already known in the fields of task arithmetic or multi-task learning.
> >
>
> Response:
>
> Thank you for the opportunity to address this concern.
>
> The reviewer suggests our key finding about the distinct roles of representation modules versus task-specific components is trivial. However, this characterization overlooks several important aspects:
> 1. While previous studies discuss weight disentanglement, they haven't explored its emergence from specific architectural components. Our work reveals that weight disentanglement isn't merely a global model property but is significantly influenced by how we fine-tune individual components.
> 2. Our ablation studies in Section 4.3 demonstrate this relationship was not predictable a priori. Previous works in task arithmetic treated model components uniformly, without distinguishing their roles.
> 3. Building on established Neural Tangent Kernel (NTK) theory, we discovered the counter-intuitive finding that task-specific components can constrain task arithmetic effectiveness, despite representation modules supporting weight disentanglement. This emerged only through systematic empirical investigation.
> 4. Our findings have both theoretical significance and practical impact, as demonstrated by improved task arithmetic performance (Tables 2-3) and new research directions in understanding component-specific roles in multi-task learning.

---

> ### Author Response · Authors · 2024-11-21
> **Rebuttal (Part 2)**
>
> > 3. The experiments are quite limited, as many related works cited by the authors are missing from the experimental results, particularly in Tables 2 and 3. This omission reduces the validity of the proposed method.
> >
>
> Response:
>
> Thank you for your feedback. We appreciate the opportunity to address your concerns regarding the experimental comparisons in our paper. We would like to offer the following rebuttal.
>
> We would like to clarify the scope and methodology of our experimental comparisons. Our work specifically addresses task arithmetic through a pre-merging approach, focusing on fine-tuning independent models (similar to Ortiz-Jimenez et al. 2024's linearized model fine-tuning). The related works mentioned in our paper fall into two distinct categories:
>
> 1. Post-merging methods (Matena & Raffel 2021; Ainsworth et al. 2023; Stoica et al. 2023; Yang et al. 2023)
> 2. Broader multi-task/transfer learning approaches (Liu et al. 2024; Ilharco et al. 2022)
>
> Neither category directly addresses task arithmetic in the pre-merging context, making them unsuitable for direct experimental comparison. Including these methods would not provide meaningful evaluation of our specific contributions. Our updated related work section elaborates on these methodological distinctions. The comprehensive evaluation within our specific domain demonstrates the validity and effectiveness of our approach.
>
> > 4. The paper's theoretical contributions rely heavily on [2], which limits its originality and overall contribution.
> >
>
> Response:
>
> Thank you for the opportunity to address this concern.
>
> While [2] provides valuable theoretical foundations in task arithmetic through NTK theory, our work takes a different direction by focusing on empirical validation and practical insights. Rather than extending theoretical frameworks, we leverage existing theory as motivation to investigate the distinct roles of different model components in task arithmetic - an aspect not previously explored.
>
> Our novel empirical findings about how representation modules and task-specific components differently affect task arithmetic effectiveness have important implications for both theory and practice. Through systematic experimentation (Section 4.3), we discovered that the kernel behavior within attention modules plays a crucial role in weight disentanglement, leading to practical improvements in task arithmetic performance (Tables 2-3).
>
> These insights, while motivated by theoretical understanding from prior work, represent significant empirical contributions that advance our understanding of task arithmetic mechanisms and open new research directions in multi-task learning.
>
> [1] Multitask prompted training enables zero-shot task generalization (ICLR 2023) [2] Task arithmetic in the tangent space: Improved editing of pre-trained models (NeurlPS 2024)

---

### Official Review · Reviewer_S2KF · 2024-11-02

**Soundness:** 3
**Presentation:** 3
**Contribution:** 3
**Rating:** 8
**Confidence:** 3

**Summary:**

This paper presents a method that fine-tunes only the attention modules of models to enhance weight disentanglement in task arithmetic. Task arithmetic is a training-free approach that enables new tasks by linearly combining the weights of a task-specific model with those of the original model. Building on NTK studies on kernel behaviour, the proposed method introduces an approximate kernel behaviour test to verify that attention modules exhibit this characteristic. The authors validate their hypothesis by fine-tuning only the attention modules in CLIP models, showing that this approach outperforms other fine-tuning strategies as well as the original NTK-initialization method.

**Strengths:**

The paper is well-organized and clearly written. The authors thoughtfully introduce relevant prior work, providing clear definitions and context for task arithmetic, weight disentanglement, and their associated challenges. Each argument is well-supported, with compelling visualizations that reinforce the method's performance. The progression from motivation to methodology is logical and engaging.

**Weaknesses:**

The paper has no prominent weaknesses. Here are some other open questions:

1. Contribution of Individual Attention Modules.
While the method targets attention modules to achieve weight disentanglement, it remains unclear whether each module contributes equally to task performance. In large models, which may have over 100 billion parameters, understanding the relative contribution of each attention module could make the approach more efficient, possibly allowing for sparse fine-tuning of only the most impactful modules. An analysis or ablation study on the importance of individual modules could provide further insights into how weight disentanglement varies within the attention layers.

2. Generalizability to Non-Attention-Based Architectures.
While the method demonstrates promising results on attention-based models, it would be useful to discuss the potential for generalizing this approach to non-attention architectures, such as convolutional or Mamba-based models. Would the same strategy apply effectively, or would alternative fine-tuning techniques be required for architectures without attention mechanisms? Providing an outlook on this point could help extend the paper's relevance across a wider variety of model architectures.

**Questions:**

See the weaknesses.

---

> ### Author Response · Authors · 2024-11-21
> **Rebuttal**
>
> Thank you for your positive feedback and insightful questions. We appreciate the opportunity to clarify and expand on our work. We are pleased that the reviewer found no prominent weaknesses in our paper. To address the open questions raised:
>
> 1. **Contribution of Individual Attention Modules:**
> We agree that understanding the relative contributions of individual attention modules is an important direction for future work. Our current study focused on demonstrating the effectiveness of fine-tuning all attention modules collectively, as this approach already showed significant improvements in weight disentanglement and efficiency compared to existing methods.
>
> In our experiments, we fine-tuned the query, key, value, and output projection matrices (Wq, Wk, Wv, Wo) in all attention layers. This approach was motivated by our observation that attention modules exhibit stronger kernel behavior compared to the full model (as shown in Figure 3). While we did not conduct an ablation study on individual modules, our results consistently showed improvements across different model sizes (ViT-B/32, ViT-B/16, ViT-L/14), suggesting that this approach is effective across various scales.
>
> We agree that exploring sparse fine-tuning of the most impactful modules could be a valuable extension of our work. This could potentially further reduce computational costs while maintaining or even improving performance. **We actually had a similar discussion in the Discussion section highlighting this as an important direction for future research: ‘Future studies could explore integrating task arithmetic with partial fine-tuning techniques, focusing on varying numbers of attention blocks. Additionally, investigating the relationship between the sparsity of attention modules and their kernel behavior may provide insights into learnable tasks.’**
>
> 2. **Generalizability to Non-Attention-Based Architectures:**
> We appreciate this insightful question about the generalizability of our approach. Our current work focused on attention-based models, specifically Vision Transformers, as they are widely used in state-of-the-art vision and language tasks. However, we agree that exploring the applicability of our method to other architectures is an important consideration.
>
> For non-attention architectures like convolutional neural networks (CNNs) or more recent models like Mamba, the core principle of our approach - identifying and fine-tuning modules that exhibit kernel behavior - could potentially be adapted. In CNNs, for instance, one might investigate whether certain convolutional layers exhibit similar properties to attention modules in terms of weight disentanglement. For Mamba models, which use selective state spaces instead of attention, a similar analysis could be conducted on their key components.
>
> We acknowledge that directly applying our method to these architectures may not be straightforward, and architecture-specific adaptations would likely be necessary. **We have had a paragraph in the discussion section addressing this limitation and suggesting it as an important area for future research: ‘Current task vectors are constrained to models with identical architectures and initializations due to their reliance on element-wise weight operations.’** This extension could indeed broaden the applicability of our approach across a wider range of model architectures.
>
> We thank the reviewer for these valuable suggestions, which have helped us to better contextualize our work and identify important directions for future research. We believe addressing these points strengthens our paper by acknowledging its current limitations and outlining potential avenues for expanding its impact.

---

> > ### Comment · Reviewer_S2KF · 2024-11-26
> >
> > Thanks for the response. After reading other reviewers' comments, I think we have a clear consensus on the contribution of this work and I would like to keep my original rating as accept.

---

### Author Response · Authors · 2024-11-21
**General Response**

We would like to express our gratitude for the insightful feedback provided by the reviewers regarding our manuscript. We appreciate the thorough evaluations and constructive suggestions, which will help us enhance the quality of our work. Below, we outline our responses to the key points raised and the revisions we have implemented.

1. **Revision of Related Work (Raised by 6SAc, Rsur)**: We incorporate references to relevant surveys and studies, such as those by Yadav et al. (2024) and Yang et al. (2024), to provide a more comprehensive overview of the current landscape in model merging techniques. Existing model merging techniques can be broadly categorized into two main types (Yang et al., 2024): (i) Pre-Merging Methods: These methods focus on enhancing the conditions necessary for effective model merging by optimizing the fine-tuning process of individual models. (ii) During Merging Methods: These approaches address task conflicts and interference through various strategies before executing the parameter merging operations.

    This Modification will clarify how our work fits into the broader context of existing research.

2. **Inclusion of NLP Results (Raised by Rsur)**: We recognize that the results for NLP tasks presented in Table 4 should be included in the main text to ensure a complete understanding of our findings. We have revised the manuscript to integrate these results, highlighting their significance alongside the vision tasks.
3. **Ablation Study in Main Text (Raised by Rsur)**: We appreciate the suggestion to mention the ablation study detailed in Appendix B.4 within the main text. We have included a summary of this study in the relevant section, emphasizing its importance in demonstrating the robustness of our approach and providing additional insights into parameter sensitivity.
4. **Parameter Sensitivity Analysis (Raised by 6SAc):** We have also included a parameter sensitivity analysis of the coefficient $\alpha$ in the main text. This analysis evaluates how different choices of $\alpha$ impact model performance and weight disentanglement. The results demonstrate the robustness of our method across a range of $\alpha$ values, further validating our hypothesis regarding the importance of careful parameter selection in optimizing task arithmetic performance.
5. **Addressing Typos and Minor Errors (Raised by Rsur)**: We conducted a thorough review of the manuscript to correct any typos or minor mistakes, enhancing overall clarity and professionalism.

We believe these revisions will address the reviewers' concerns effectively and strengthen our manuscript significantly. Thank you once again for your valuable feedback, and we’re happy to address any further questions.

Yadav et al. A Survey on Model MoErging: Recycling and Routing Among Specialized Experts for Collaborative Learning.

Yang et al. Model Merging in LLMs, MLLMs, and Beyond: Methods, Theories, Applications and Opportunities.

---

### Author Response · Authors · 2024-11-24
**Follow up on Rebuttal**

Dear Reviewers,

We sincerely appreciate the time and effort you have dedicated to reviewing our work.

We have thoroughly addressed all the concerns raised, including the related work highlighted by Reviewer 6SAc and the parameter sensitivity analysis noted by Reviewer 6SAc. Additionally, we have incorporated the NLP and ablation results into the main text, as suggested by Reviewer Rsur. We have also made significant efforts to clarify our method and contributions, as emphasized by Reviewer CQUv.

As the discussion period comes to a close, we want to ensure that our responses have fully addressed your questions and concerns. Please let us know if there are any aspects of our work that still require clarification or further discussion.

Thank you once again for your valuable feedback.

Best regards,

The Authors

---

### Meta-Review · Area_Chair_kLwv · 2024-12-20

**Metareview:**

Summary
The paper studies task disentanglement in neural networks whereby new tasks can be solved by linearly combining weights of task-specific models. The authors build upon NTK and show that finetuning only the attention modules of a model is amenable to task disentanglement. The method is verified on CLIP models and outperforms finetuning strategies.

Strengths
1. The authors study task disentanglement and show that attention modules can be finetuned but still retain a linear entanglement property.
2. The experimental setup in this work studies different sized models on image classification and NLP tasks.
3. The paper is presented and written well

Weaknesses
1. The approach requires tuning a hyperparameter \alpha and is sensitive to its value as the author rebuttal also shows. An automatic tuning approach for this would make the work more impactful.
2. The approach is only applicable to models with an attention layer.

Justification
The paper received a majority of positive reviews. The author rebuttal answered the concerns from the reviewers. The paper is technically solid and presents an interesting approach for task disentanglement.

**Additional Comments On Reviewer Discussion:**

One reviewer gave this paper a negative review but their concerns were addressed in the rebuttal and the reviewer didn't respond. The other reviewer concerns were addressed by the authors as well.

---

### Decision · Program_Chairs · 2025-01-22

Accept (Poster)